

# The dopamine receptor $D_5$ gene shows signs of independent erosion in toothed and baleen whales

Luís Q. Alves[1,2], Juliana Alves[1,2], Rodrigo Ribeiro[1,2], Raquel Ruivo[1] and Filipe Castro[1]

[1] CIIMAR-University of Porto, Matosinhos, Portugal
[2] FCUP-University of Porto, Porto, Portugal

## ABSTRACT

To compare gene loci considering a phylogenetic framework is a promising approach to uncover the genetic basis of human diseases. Imbalance of dopaminergic systems is suspected to underlie some emerging neurological disorders. The physiological functions of dopamine are transduced via G-protein-coupled receptors, including $DRD_5$ which displays a relatively higher affinity toward dopamine. Importantly, $DRD_5$ knockout mice are hypertense, a condition emerging from an increase in sympathetic tone. We investigated the evolution of $DRD_5$, a high affinity receptor for dopamine, in mammals. Surprisingly, among 124 investigated mammalian genomes, we found that Cetacea lineages (Mysticeti and Odontoceti) have independently lost this gene, as well as the burrowing *Chrysochloris asiatica* (Cape golden mole). We suggest that $DRD_5$ inactivation parallels hypoxia-induced adaptations, such as peripheral vasoconstriction required for deep-diving in Cetacea, in accordance with the convergent evolution of vasoconstrictor genes in hypoxia-exposed animals. Our findings indicate that Cetacea are natural knockouts for $DRD_5$ and might offer valuable insights into the mechanisms of some forms of vasoconstriction responses and hypertension in humans.

## INTRODUCTION

Dopamine is a neurotransmitter essential for brain function, regulating various physiological processes including locomotion, cognition, and neuroendocrine functions (*Hollon et al., 2002*; *Ott & Nieder, 2019*). Dopamine molecular actions are transduced *via* a specific group of G-protein coupled receptors entailing two major classes: $DRD_1$-like and $DRD_2$-like receptors (*Beaulieu & Gainetdinov, 2011*; *Opazo et al., 2018*). While $DRD_1$-like receptors stimulate cAMP production postsynaptically, $DRD_2$-like receptors inhibit cAMP production both pre and postsynaptically (*Beaulieu & Gainetdinov, 2011*). The genomic structure of the underlying genes is also distinct, with $DRD_1$-like receptors yielding single exon coding regions (*Beaulieu & Gainetdinov, 2011*). $DRD_{1-2}$ receptor classes have diversified in vertebrate evolution most likely as a result of genome duplications (*Opazo et al., 2018*). Interestingly, agonist and antagonist amino acid site conservation suggests evolutionary stasis of dopaminergic pathways (*Opazo et al., 2018*). Among $DRD_1$-like

Corresponding authors
Raquel Ruivo, rruivo@ciimar.up.pt
Filipe Castro,
filipe.castro@ciimar.up.pt

receptors, the $DRD_5$ subtype displays distinctive features, namely a relatively higher affinity toward dopamine, a putative agonist-independent activity and low level, yet widespread, brain expression (*Beaulieu & Gainetdinov, 2011*; *Ciliax et al., 2000*; *Sunahara et al., 1991*; *Tiberi & Caron, 1994*). Nonetheless, the $DRD_5$ seems to display distinct regional and cellular distribution patterns in the brain, when compared to the $DRD_1$ and $DRD_2$ subtypes, with protein enrichment detected in the cerebral cortex, hippocampus and basal ganglia (*Ariano et al., 1997*; *Ciliax et al., 2000*). Peripheral expression has also been found in the adrenals (*Dahmer & Senogles, 1996*), kidney (*Sanada et al., 2000*) and gastrointestinal tract (*Mezey et al., 1996*). Despite the association with schizophrenia (*Muir et al., 2001*), attention-deficit/hyperactive disorder (*Daly et al., 1999*) and substance abuse (*Vanyukov et al., 1998*), gene targeting studies revealed that $DRD_5$ knock-out mice develop hypertension, showing increased blood pressure from 3 months of age (*Hollon et al., 2002*). This hypertensive phenotype appears to result from a central nervous system defect, leading to an increase in sympathetic tone and, consequently, vasoconstriction (*Hollon et al., 2002*). Besides neuronal impairment, $DRD_5$ disruption was also suggested to increase the expression of the prohypertensive Angiotensin II Type 1 Receptor ($AT_1R$), involved in renal salt balance, blood pressure, and vasoconstriction (*Li et al., 2008*). In fact, renal dopamine was suggested to promote salt excretion, counter-regulating $AT_1R$, and lowering hypertensive states (*Hong et al., 2017*; *Li et al., 2008*; *Zeng et al., 2005*).

Whole genome sequencing has greatly expanded our capacity to comprehend evolutionary history, the role of adaptation or the basis for phenotype differences across the tree of life. Multi-genome comparisons have also been powerful to recognize the molecular basis of human diseases, a field named as phylomedicine (*Emerling et al., 2017*; *Kumar et al., 2011*; *Miller & Kumar, 2001*; *Springer & Murphy, 2007*). Here, we investigate the evolution of $DRD_5$ in mammalian species. By analyzing 124 genomes covering 16 orders, we show that independent coding-debilitating mutations occurred in the ancestors of Mysticeti, of *Physeter macrocephalus* (sperm whale) and the remaining Odontoceti, strongly suggesting that $DRD_5$ is non-coding in Cetacea. Reductive episodes have been widely documented across the tree of life contributing to organismal divergence and physiological and morphological adaptation to environmental cues (*Albalat & Cañestro, 2016*; *Braun, 2003*; *Jeffery, 2009*; *Olson, 1999*). In agreement, gene loss mechanisms seem pervasive in lineages that endured drastic habitat transitions in the course of evolution, such as Cetacea, entailing niche-specific adaptations (*Huelsmann et al., 2019*; *Lachner et al., 2017*; *Lopes-Marques et al., 2018*; *Lopes-Marques et al., 2019a*; *McGowen, Gatesy & Wildman, 2014*; *Nery, Arroyo & Opazo, 2014*; *Sharma et al., 2018*; *Strasser et al., 2015*). Thus, our findings suggest that these species are natural KOs for this dopamine receptor and might offer valuable insights into the mechanisms of some forms of essential hypertension.

## MATERIALS AND METHODS

To manually infer the coding status of $DRD_5$ genes, the following strategy was used: first, gene orthology was assessed via synteny analysis, to clarify cases where gene annotation was not found, as well as to define genomic regions to be posteriorly collected for gene annotation (see Fig. S1). Next, we performed the manual annotation of the open reading

frame (ORF), and the correspondent sequence was screened for abolishing mutations. At least one mutation was posteriorly validated using raw genomic sequencing reads available at the National Center of Biotechnology Information (NCBI) sequence read archive (SRA) database. Finally, $d_N/d_S$ analyses were also conducted to further investigate the inactive status of $DRD_5$ in Cetacea (see below).

## Synteny analysis

To build the synteny maps for the $DRD_5$ gene *locus* in Cetacea and *Hippopotamus amphibius* (common hippopotamus) several annotated Cetacea genome assemblies were inspected and scrutinized using the NCBI browser, namely *Orcinus orca* (killer whale; GCF_000331955.2), *Lagenorhynchus obliquidens* (Pacific white-sided dolphin; GCF_003676395.1), *Tursiops truncatus* (common bottlenose dolphin; GCF_001922835.1), *Delphinapterus leucas* (beluga whale; GCF_002288925.1), *Neophocaena asiaeorientalis asiaeorientalis* (Yangtze finless porpoise; GCF_003031525.1), *Lipotes vexillifer* (Yangtze River dolphin; GCF_000442215.1), *Physeter macrocephalus* (sperm whale; GCF_002837175.1) and *Balaenoptera acutorostrata scammoni* (minke whale; GCF_000493695.1). *Bos taurus* (cattle; GCF_002263795.1), a fully terrestrial relative of extant cetaceans, was used as reference. Next, (1) in genome assemblies with annotated $DRD_5$, the following procedure was used: five protein-coding genes, upstream and downstream of the $DRD_5$ gene, and from the same strand, were collected; (2) if $DRD_5$ gene annotations were not present, the genomic *locus* was retrieved using *Bos taurus* (cattle) $DRD_5$ flanking genes as reference. The selected anchoring genes to search upstream and downstream $DRD_5$ neighboring genes were CPEB2 and OTOP1, respectively. Regarding *Hippopotamus amphibius* (common hippopotamus), the synteny map was built via BLAST searches against the assembled, fragmented and unannotated genome of the same species, available at NCBI (GCA_002995585.1). *Bos taurus* (cattle) $DRD_5$ flanking genes were used as reference; using the discontiguous megablast task from blastn, the best BLAST hit (highest alignment identity and query coverage) was retrieved and the coordinates of the alignment in the target genome carefully inspected. The *Hippopotamus amphibius* (common hippopotamus) synteny map was then built by sorting the genes according to the subject alignment coordinates within genes aligning at the same genomic scaffold.

## Sequence retrieval and gene annotation

The NCBI "low-quality protein" (LQ) tag marks RefSeq sequences that have been automatically modified relative to the corresponding genome sequence to correct for possible ORF protein-altering indels or mismatches, assuming that these arise from sequencing errors or genome assembly artefacts. Yet, in several cases these frameshift and nonsense codons correspond to authentic biological modifications leading to shifts or truncations of the predicted ORF, culminating into an abnormal protein amino acid constitution. For this reason, to clarify the functional status of $DRD_5$ in species presenting LQ tag for this gene, the corresponding genomic regions were directly collected from NCBI.

Concerning species with no annotated DRD$_5$ genes two scenarios were identified: (1) annotated genomes excluding DRD$_5$ gene annotation and (2) unannotated genomes. Regarding the first case (i.e., the cetaceans *Lipotes vexillifer*, the Yangtze River dolphin and *Balaenoptera acutorostrata scammoni*, the minke whale, as well as other non-cetacean mammals), the DRD$_5$ genomic *locus* was retrieved using, as reference, annotated DRD$_5$ flanking genes from closely related species (for cetaceans, the terrestrial extant sister clade *Bos taurus* (cattle), regarding non-cetacean mammals see Table S1). If the genomic *locus* exhibited severe genomic fragmentation (presence of Ns), thus, hindering the retrieval using neighboring genes, blastn searches were conducted against the Whole Shotgun Contigs of the corresponding species via discontiguous megablast task, using as query the DRD$_5$ coding sequence (CDS) of the same reference species (Table S1). The genomic sequence corresponding to the BLAST hit with the highest alignment identity and query coverage was selected.

For species without annotated genomes (i.e., *Balaenoptera bonaerensis* (Antarctic minke whale), *Eschrichtius robustus* (gray whale), *Balaena mysticetus* (bowhead whale), *Sousa chinensis* (Indo-pacific humpbacked dolphin), as well as *Hippopotamus amphibius* (hippopotamus)) genomic sequences were retrieved through blastn searches in the corresponding genome assembly using the *Bos taurus* (cattle) DRD$_5$ CDS as query. For each species, the best genomic scaffold corresponded to the BLAST hit with the highest query coverage and identity value. Due to the presence of a fragmented genomic region in the DRD$_5$ gene annotation of *T. truncatus* (common bottlenose dolphin), the same BLAST search procedure was carried out for this species, using as target the Whole Genome Shotgun contig dataset of the same species.

Collected genomic sequences were further imported into Geneious Prime 2019 (www.geneious.com) and the DRD$_5$ gene CDSs manually annotated for each species. Briefly, using the built-in map to reference tool with the highest sensibility parameter selected, the reference single-exon DRD$_5$ gene, 3′ and 5′ UTR flanked, was mapped against the corresponding genomic sequence of the in-study species. Aligned regions were further carefully screened for ORF abolishing mutations including frameshift mutations and in-frame premature stop codons. For Cetacea and *Hippopotamus amphibius* (common hippopotamus) DRD$_5$ gene annotation, *Bos taurus* (cattle) DRD$_5$ was selected as reference. Regarding non-cetacean mammals DRD$_5$ annotation, different references were chosen according to the phylogenetic relationships between reference and test species (Table S1). Finally, the identified mutations were next validated using raw sequencing reads retrieved from at least two independent genomic NCBI SRA projects (when available). Briefly, blastn searches were conducted against the selected SRA projects (Supplemental Materials 1–3) using as query the nucleotide sequence containing the mutation. BLAST hits were imported into Geneious Prime 2019 and mapped against the manually annotated sequences, using the built-in map to reference tool, to confirm the presence of the identified mutation.

## Phylogenetic and $d_N/d_S$ analyses

The predicted cetacean and *Hippopotamus amphibius* (common hippopotamus) DRD$_5$ CDSs, as well as the DRD$_5$ CDSs of *Homo sapiens* (human) and *Bos taurus* (cattle)

**Table 1  Selection ($d_N/d_S$) analyses with CODEML from PAML for the seven different branch categories.**

| Branch category | $d_N/d_S$ ($\omega$) |
|---|---|
| Functional branches<br>(*Homo sapiens, Bos taurus, Hipoppotamus amphibius*) | 0.09 (*p*= **0.030**) |
| Mysticeti branches<br>(*Balaena mysticetus, Eschrichtius robustus, Balaenoptera bonaerensis, Balaenoptera acutorostrata scammoni*) | 0.460 (*p*= 0.291) |
| Odontoceti branches<br>(*Physeter macrocephalus, Lipotes vexillifer, Orcinus orca, Lagenorhynchus obliquidens, Tursiops truncatus, Sousa chinensis, Neophocanea asiaeorientalis asiaeorientalis*) | 0.386 (*p*= 0.692) |
| Common cetacean branch | 0.135 (*p*= **0.001**) |
| Stem Mysticeti branch | 1.174 (*p* = 0.905) |
| Stem Odontoceti (excluding *Physeter macrocephalus*, sperm whale) branch | 0.213 (*p*= **0.045**) |
| *Physeter macrocephalus* (sperm whale) ancestor branch | 0.320 (*p*= 0.460) |

**Notes:**
Likelihood ratio tests (LRT) corresponding *p*-value concerning each branch category is also presented.
$d_N/d_S$ values are significantly different from 1 if the correspondent LRT *p*-value < 0.05 (for a 95% confidence level, in bold).

available at NCBI were translation aligned in Geneious Prime 2019 using the Blosum62 substitution matrix. The alignment (1,440 bp) was manually inspected and predicted $DRD_5$ CDSs involved in alignment suffered prior inspection, with frameshift insertions and deletions being omitted and stop codons recorded as missing. The alignment was posteriorly exported for phylogenetic analysis. A Maximum likelihood phylogenetic tree, performed in PhyML3.0 server (*Guindon et al., 2010*), was produced with the best sequence evolutionary model being determined using the built-in smart model selection (HKY85 +G) (*Lefort, Longueville & Gascuel, 2017*). Node support was inferred based on 1,000 bootstrap replicates. The resulting phylogenetic tree newick file is available as File S1 and was subsequently imported for visualisation in FigTree (Supplemental Material 4) (*Rambaut & Drummond, 2012*).

PAML 4.6 (*Yang, 2007*) was used to estimate the ratio ($\omega$) of the nonsynonymous substitution rate ($d_N$) to the synonymous substitution rate ($d_S$) using the produced phylogenetic tree (Supplemental Material 4). Codeml with the branch model was implemented to estimate $d_N/d_S$ ratios ($\omega$) for seven different branch categories: Mysticeti, Odontoceti, Functional, a category corresponding to the common cetacean branch, other concerning the stem Mysticeti branch, an additional one regarding the stem Odontoceti branch (excluding *Physeter macrocephalus*, the sperm whale) and finally, a category corresponding to the *Physeter macrocephalus* (sperm whale) ancestor branch (Table 1). Mysticeti and Odontoceti branch categories comprised all predicted Mysticeti and Odontoceti $DRD_5$ sequences, respectively. The Functional branch category comprised *Hippopotamus amphibius* (common hippopotamus), *Bos taurus* (cattle) and *Homo sapiens* (human) $DRD_5$ CDSs. For each of the seven different branch categories, likelihood ratio tests (following $\chi^2$ distribution) were conducted to compare the estimated ratio ($\omega$) of the nonsynonymous substitution rate ($d_N$) to the synonymous substitution rate ($d_S$) determined by PAML (the alternative hypothesis), with the expected ratio ($\omega$) of the nonsynonymous substitution rate ($d_N$) to the synonymous substitution rate ($d_S$) according to a neutral model of evolution ($\omega = 1$) (the null hypothesis). All PAML analyses were run

 

with the CodonFreq set to three and codon sites with ambiguous data (including gaps and missing data) were included in the analyses.

## RESULTS

To examine the annotation tags and distribution of the $DRD_5$ gene across mammals, 119 annotated mammalian genomes available at NCBI were scrutinized for the presence of $DRD_5$ gene annotation and each respective protein product description screened for the LQ tag. This examination resulted in 10 species presenting the $DRD_5$ LQ tag, including *Ovis aries* (sheep), *Phascolarctos cinereus* (koala), *Bison bison bison* (plains bison), *Myotis davidii* (vesper bat), *Ochotona princeps* (American pika) and five cetacean species. The latter included *Lagenorhynchus obliquidens* (Pacific white-sided dolphin), *N. a. asiaeorientalis* (Yangtze finless porpoise), *D. leucas* (beluga whale), *Physeter macrocephalus* (sperm whale) *and Orcinus orca* (killer whale). Each genomic sequence corresponding to the $DRD_5$ LQ annotations was examined and the CDS manually predicted (*Lopes-Marques et al., 2017*). Given the prominence of $DRD_5$ LQ annotations in Cetacea we scrutinized other cetacean species with available, but unannotated genomes, *Balaenoptera bonaerensis* (Antarctic minke whale), *Eschrichtius robustus* (gray whale), *Balaena mysticetus* (bowhead whale), *S. chinensis* (Indo-Pacific humpback dolphin), or with annotated genomes lacking $DRD_5$ annotations: *Lipotes vexillifer* (Yangtze River dolphin) and *Balaenoptera acutorostrata scammoni* (minke whale) (Fig. S1). Additionally, *T. truncatus* (bottlenose dolphin), presenting a seemingly intact $DRD_5$ gene annotation, without the LQ tag, as well as *Hippopotamus amphibius* (common hippopotamus) predicted $DRD_5$ CDS, representing the closest extant lineage of Cetacea, were equally inspected. Other mammals with annotated genome without $DRD_5$ gene annotation were also scrutinized, namely: *Microcebus murinus* (gray mouse lemur), *Jaculus jaculus* (lesser Egyptian jerboa), *Chrysochloris asiatica* (Cape golden mole), *Erinaceus europaeus* (western European hedgehog), *Elephantulus edwardii* (Cape elephant shrew) and *Condylura cristata* (star-nosed mole). In total, 124 mammalian species were inspected and an in-depth description regarding analyzed species list and genomic sequences accession numbers are available at Table S2. Figure S2 presents a multiple translation alignment of the NCBI non-LQ tagged mammalian $DRD_5$ orthologous sequences. The alignment also includes the predicted *Hippopotamus amphibius* (hippopotamus) $DRD_5$ sequence and excludes *T. truncatus* (common bottlenose dolphin) $DRD_5$ sequence, afterward demonstrated to contain inactivating mutations (see below). The examined sequences exhibit a substantial degree of conservation (average pairwise identity of over 80%), with minor variation in the expected protein size. The protein sequence conservation is particularly noticeable at the c-terminus (Fig. S2).

### ORF-disrupting mutations of $DRD_5$ in Cetacea

For cetacean species with annotated genomes, we started by examining the $DRD_5$ gene *locus*, including neighboring genes, to verify and elucidate the orthology of the annotated and non-annotated genes and outline the genomic regions to be inspected (Fig. S1). All analyzed loci were found to be conserved, including in both *Lipotes vexillifer* (Yangtze

River dolphin) and *Balaenoptera acutorostrata scammoni* (minke whale), which lacked previous DRD$_5$ gene annotations (Fig. S1). Subsequent manual annotation of all collected cetacean genomic sequences revealed DRD$_5$ gene erosion across all analyzed species, except in *Balaenoptera acutorostrata scammoni* (minke whale), for which the DRD$_5$ coding status could not be accessed due to fragmentation of the 5′ end of the respective genomic region (presence of sequencing gaps (Ns)). In detail, a conserved 2-nucleotide deletion was detected for the full set of Odontoceti examined species, except for *Physeter macrocephalus* (sperm whale) that presented a premature stop codon near the middle of the gene and a single nucleotide insertion close to the end of the gene (Fig. 1). The 2-nucleotide deletion alters the reading frame, leading to a drastic change in downstream amino acid composition. Additionally, non-conserved mutations were found in *Lipotes vexillifer* (Yangtze River dolphin), which presented two premature stop codons, *N. a. asiaeorientalis* (Yangtze finless porpoise) that presented a premature stop codon close to the middle of the ORF, and *D. leucas* (beluga whale) with a single nucleotide deletion near the 5′ end of the gene (Fig. 1). *D. leucas* (beluga whale) presumed DRD$_5$ sequence presented another noticeable feature. A massive and abrupt alignment identity decrease observed when aligning *Bos taurus* (cattle) DRD$_5$ gene against the genomic target region of *D. leucas* (beluga whale). The alignment identity drop was noted approximately in the middle of the complete alignment length for this species, suggesting that the DRD$_5$ gene sequence is interrupted, and further supporting pseudogenization in this species. Regarding Odontoceti, at least one ORF-abolishing mutation was validated using available genomic SRAs experiments for all studied species, excluding *S. chinensis* (Indo-pacific humpbacked dolphin) and *Lipotes vexillifer* (Yangtze River dolphin) for which no genomic sequencing runs were available at the NCBI SRA database (Supplemental Material 1).

Regarding the Mysticeti suborder, a conserved single nucleotide deletion was detected in all species except *Balaenoptera acutorostrata scammoni* (minke whale) (Fig. 1). A non-conserved 2-nucleotide insertion was found also found in *Balaenoptera bonaerensis* (Antartic minke whale) and an insertion of one nucleotide was detected at *Eschrichtius robustus* (gray whale) near the 5′ end of the DRD$_5$ sequence (Fig. 1). Again, at least one ORF-abolishing mutation was validated using genomic SRA experiments for all analyzed species (Supplemental Material 1). Importantly, no conserved mutations were detected between most Odontoceti, *Physeter macrocephalus* (sperm whale, Odontoceti) and Mysticeti lineages, suggesting that three pseudogenization events occurred independently after their evolutionary divergence (Fig. 1). To increase the robustness of our analysis, we further scrutinized the genome of the extant sister clade of the Cetacea, the Hippopotamidae and were able to predict a fully functional CDS for DRD$_5$ in *Hippopotamus amphibius* (common hippopotamus), supporting the loss of DRD$_5$ after Cetacea diversification.

## $d_N/d_S$ analyses support independent inactivation events of DRD$_5$ in Cetacea lineages

To further strengthen the mutational analyses, we next carried out a $d_N/d_S$ approach. The $d_N/d_S$ analyses for species with functional DRD$_5$ gene (Functional branches category)

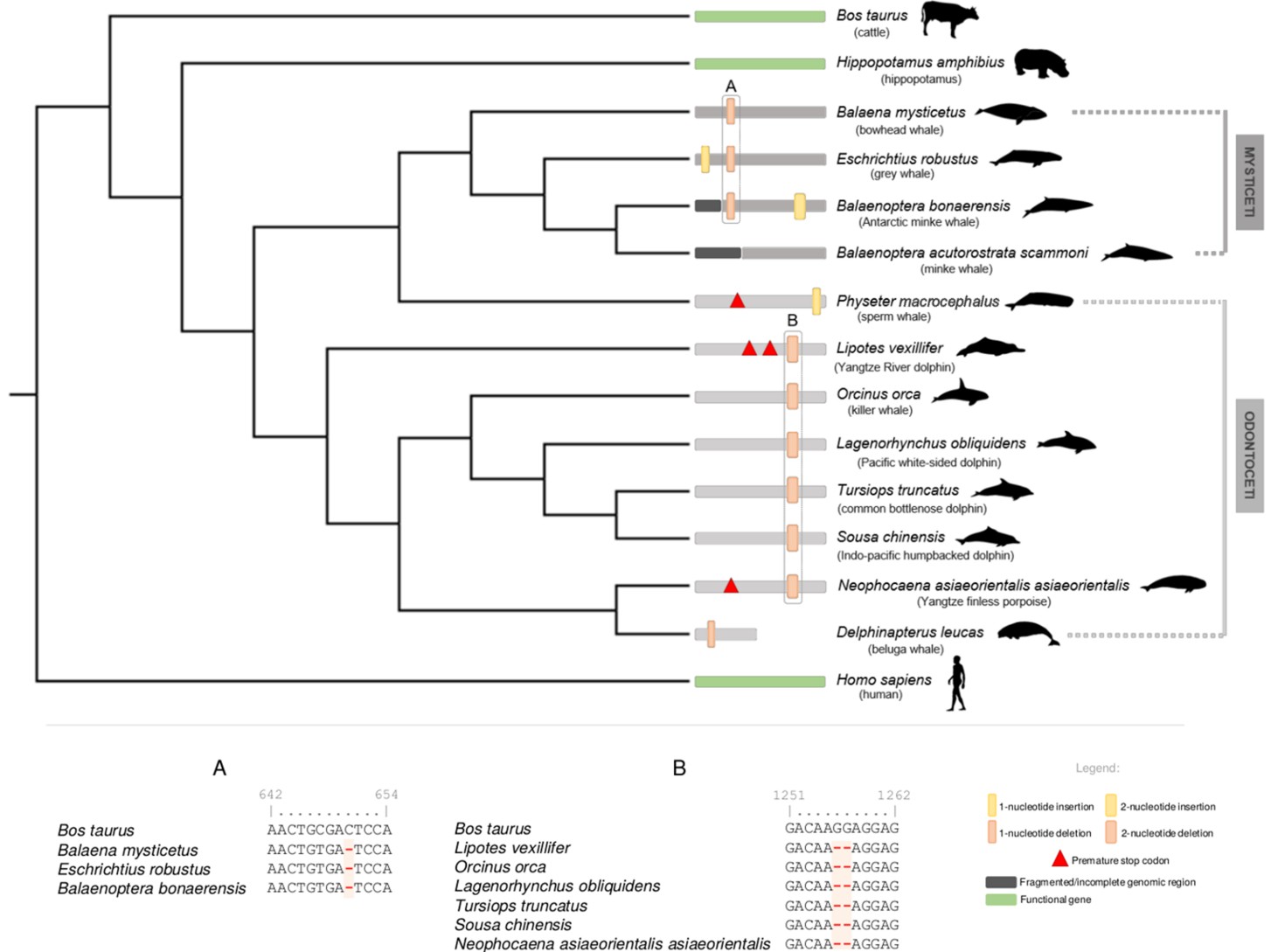

**Figure 1 Schematic representation of the annotated DRD₅ gene ORF-abolishing mutations regarding Cetacea parvorders Odontoceti and Mysticeti.** Phylogenetic relationships derived from a maximum likelihood (ML) tree with the DRD₅ sequences from Mysticeti and Odontoceti species, as well as from *Hippopotamus amphibius* (hippopotamus), *Bos taurus* (cattle) and *Homo sapiens* (human). DRD₅ ORF-abolishing mutations are mapped in the corresponding branch. Vertical thick bars represent 2-nucleotide insertions (yellow) or deletions (orange). Vertical thin bars represent single nucleotide insertion (yellow) or deletion (orange). In-frame premature stop codons are represented by red triangles. Fragmented or incomplete genomic regions are represented by black regions. *Delphinapterus leucas* (beluga whale) predicted DRD₅ truncation is represented by a smaller bar. Functional branches are represented by green bars, nonfunctional branches are represented by gray and dark gray bars (Odontoceti and Mysticeti branches, respectively). Example of DRD₅ open reading frame (ORF) inactivating mutations concerning Mysticeti clade (A) and Odontoceti clade (excluding *Physeter catodon*, the sperm whale) (B). Numbers above characters represent the alignment position index. Silhouettes were sourced from Phylopic (http://phylopic.org): the Cetacea species image credits are to Chis huh (Attribution-ShareAlike 3.0 Unported (CC BY-SA 3.0).

rejected the null hypothesis ($\omega = 0.09$, $p = 0.030$, <0.05), confirming that, as expected, these have evolved under purifying selection ($\omega < 1$), favoring the conservation of DRD₅ in the tested lineages (Table 1). In contrast, concerning Mysticeti and Odontoceti branch categories, the performed analyses failed to reject the null hypothesis ($\omega = 0.386$, $p = 0.692$ for Odontoceti branches category and $\omega = 0.460$, $p = 0.291$ for Mysticeti branches category,

respectively) suggesting that DRD$_5$ evolved neutrally in both lineages, following gene inactivation events (Table 1). Moreover, evidences of purifying selection were found for the common Cetacean branch category ($\omega = 0.135$, $p = 0.001$), suggesting the existence of a functional DRD$_5$ in the common ancestor to all cetaceans (Table 1). Purifying selection was also detected for the Odontoceti stem branch category (excluding *Physeter macrocephalus*, the sperm whale) ($\omega = 0.213$, $p = 0.045$). In contrast the *Physeter macrocephalus* (sperm whale) ancestor branch category, does not present a $d_N/d_S$ ratio ($\omega$) statistically significantly different from 1 ($\omega = 0.320$, $p = 0.460$) (Table 1). Together with the mutational evidences (Fig. 1), these results suggest two independent inactivation events within the Odontoceti. Regarding the Mysticeti stem branch category, selection analysis suggested neutral evolution ($\omega = 1.174$, $p = 0.905$) (Table 1). However, mutational evidence supports a conserved inactivation event in the common ancestor of Mysticeti. Taken together, our results support at least two independent DRD$_5$ inactivation events within Cetacea.

## Other mammalian species displaying DRD$_5$ LQ tags have a coding gene

Initial analysis revealed the presence of at least one ORF-abolishing mutation in *Ovis aries* (sheep), *Phascolarctos cinereus* (koala), *Bison bison bison* (plains bison) and *Ochotona princeps* (American pika). These are in some cases suggestive of gene inactivation and not sequencing artefacts (*Emerling et al., 2017*; *Lopes-Marques et al., 2017*). Manual annotation, including SRA validation, unveiled sequencing reads supporting the absence of disruptive ORF mutations, rebutting each inactivation mutation and suggesting that DRD$_5$ is, in fact, coding in these species (Supplemental Material 2). Regarding *Myotis davidii* (vesper bat), the fragmentation of the genomic region (Ns) flanked by upstream and downstream DRD$_5$ neighboring genes impeded us to infer the DRD$_5$ coding status in this species. Interestingly, regarding *Ochotona princeps* (American pika) mutational SRA validation, two scenarios were observed: approximately 50% of aligned reads supported the presence of a premature stop codon in the DRD$_5$ gene of this species, with the remaining set of aligning reads supporting the absence of a premature stop codon in the same species. This suggest that a polymorphic loss event might have occurred in this species, with non-functional and functional alleles currently segregating in the correspondent population.

## *Chrysochloris asiatica* presents a non-functional DRD$_5$ gene

Next, we examined other mammalian species with annotated genome yet lacking DRD$_5$ gene annotations. Results were inconclusive regarding the coding status of *Microcebus murinus* (gray mouse lemur), *Condylura cristata* (star-nosed mole), *J. jaculus* (lesser Egyptian jerboa) and *Erinaceus europaeus* (western European hedgehog) DRD$_5$, due to the fragmentation (Ns) of the genomic region flanked by the upstream and downstream DRD$_5$ neighboring genes, and the unavailability of whole genome shotgun contigs spanning our target gene. For *Elephantulus edwardii* (Cape elephant shrew) we were able to deduce a fully functional DRD$_5$ CDS. Curiously, *Chrysochloris asiatica* (Cape golden mole) presented a single nucleotide insertion in the 5′ end of the gene (validated by genomic
SRA, see Supplemental Material 3), suggesting that $DRD_5$ might be pseudogenized in this species.

## DISCUSSION

The rise of large-scale genomic sequencing projects has emphasized the role of gene loss as a potent driver of evolutionary change: underlying phenotypic adaptations or neutral regressions in response to specific environmental cues and niches (*Albalat & Cañestro, 2016*; *Braun, 2003*; *Jebb & Hiller, 2018*; *Jeffery, 2009*; *Olson, 1999*; *Sadier et al., 2018*; *Sharma et al., 2018*; *Somorjai et al., 2018*). By comparing 124 mammalian genomes, we document three independent erosion events of a dopamine receptor, $DRD_5$, in Cetacea lineages. Although the current analysis is highly dependent of the quality of genome sequencing projects and their assembly, the conserved mutational profile of $DRD_5$ gene inactivation within Cetacea lineages further strengthens our findings.

Dopamine, a neurotransmitter and signaling molecule, is involved in distinct functions both in the central nervous system and peripheral tissues: including movement, feeding, sleep, reward, learning and memory as well as in the regulation of olfaction, hormone pathways, renal functions, immunity, sympathetic regulation, and cardiovascular functions, respectively (*Beaulieu & Gainetdinov, 2011*). More specifically, the disruption of $DRD_5$-dependent pathways in rodents was shown to increase blood pressure and sympathetic tone, promoting vasoconstriction, thus yielding a hypertensive phenotype (*Hollon et al., 2002*; *Li et al., 2008*).

The observed gene loss distribution, and predicted phenotypic outcome, is consistent with the peripheral vasoconstriction mechanism described in Cetacea, suggested to counterbalance deep-diving induced hypoxia (*Ramirez, Folkow & Blix, 2007*; *Tian et al., 2016*). In fact, Cetacea have developed a number of physiological adaptations to offset hypoxia: notably, increased blood volume and oxygen-transport protein levels (hemoglobin, neuroglobin, and myoglobin), allowing oxygen stores in blood, muscle tissues, and brain, reduced heart rate or bradycardia, apnea and peripheral vasoconstriction (*Nery, Arroyo & Opazo, 2013*; *Panneton, 2013*; *Ramirez, Folkow & Blix, 2007*; *Tian et al., 2016*). Peripheral vasoconstriction allows the regional compartmentalization of blood supplies, reducing blood flow in more hypoxia-tolerant tissues, such as skin, muscle, spleen or kidney, while maintaining arterial blood flow to the central nervous system and heart (*Panneton, 2013*). Moreover, the deviation of muscle blood supply reduces lactate accumulation (*Panneton, 2013*). Thus, in Cetacea, $DRD_5$ loss could contribute for the peripheral vasoconstriction requirements of diving. Also, by shifting renal salt balance, $DRD_5$ could also play a role in the maintenance of an adequate blood volume and pressure (*Bie, 2009*; *Hong et al., 2017*; *Li et al., 2008*; *Zeng et al., 2005*).

The predicted vasoconstriction phenotype is in agreement with a previous work reporting episodes of adaptive evolution (positive selection) in genes related with hypoxia tolerance in Cetacea, including genes involved in oxygen transport and regulation of vasoconstriction (*Tian et al., 2016*). In addition, by expanding their analysis to other non-aquatic hypoxic environments, such as underground tunnels, they uncovered convergent evolution scenarios in species adapted to diving and burrowing

(*Tian et al., 2016*). A similar convergence is observed in the present work. In fact, in the mole *Chrysochloris asiatica* (Cape golden mole), $DRD_5$ was also predicted non-functional. Although this was the single $DRD_5$ loss example found outside Cetacea, one cannot discard the putative contribution of alternative molecular events (i.e., post-translational mechanisms) toward trait loss (*Sadier et al., 2018*), or event distinct physiological adaptations to overcome oxygen deprivation.

Besides diving physiology, $DRD_5$-dependent sympathetic tone alterations could also contribute to the idiosyncratic sleep behavior observed in Cetacea (*Huelsmann et al., 2019*; *Lopes-Marques et al., 2019b*; *Lyamin et al., 2008*). Several physiological adjustments occur during sleep, encompassing thermoregulation, as well as endocrine, immune, pulmonary, and cardiovascular functions (*Giglio et al., 2007*). In most mammals, sleep states lead to a decrease of the sympathetic tone, inducing vasodilation and decreasing blood pressure (*Giglio et al., 2007*). Thus, $DRD_5$ loss could prevent sympathetic tone decrease in resting states paralleling the unihemispheric sleeping behavior and long-term vigilance observed in Cetacea.

## CONCLUSIONS

Overall, our findings provide evidence for natural occurring KO for $DRD_5$. Besides highlighting a molecular signature for vasoconstriction and blood pressure regulation in Cetacea, naturally occurring $DRD_5$ KO could also provide useful frameworks to gain insight into hypertension and heart failure-induced peripheral vasoconstriction responses in humans (*Triposkiadis et al., 2009*; *Wang et al., 2008*).

## ACKNOWLEDGEMENTS

We acknowledge the various genome consortiums for sequencing and assembling the genomes.

### Funding
This work was supported by Project No. 031342 co-financed by COMPETE 2020, Portugal 2020 and the European Union through the ERDF, and by FCT through national funds. The funders had no role in study design, data collection and analysis, decision to publish, or preparation of the manuscript.

### Grant Disclosures
The following grant information was disclosed by the authors:
Project No. 031342 co-financed by COMPETE 2020, Portugal 2020 and the European Union through the ERDF, and by FCT through national funds.

### Competing Interests
The authors declare that they have no competing interests.
## Author Contributions

- Luís Q. Alves performed the experiments, analyzed the data, prepared figures and/or tables, authored or reviewed drafts of the paper, approved the final draft.
- Juliana Alves performed the experiments, analyzed the data, approved the final draft.
- Rodrigo Ribeiro performed the experiments, analyzed the data, approved the final draft.
- Raquel Ruivo conceived and designed the experiments, analyzed the data, prepared figures and/or tables, authored or reviewed drafts of the paper, approved the final draft.
- Filipe Castro conceived and designed the experiments, analyzed the data, authored or reviewed drafts of the paper, approved the final draft.

## Data Availability

    Raw data are available in the Supplemental Files.

## Supplemental Information

Supplemental information for this article can be found online at http://dx.doi.org/10.7717/peerj.7758#supplemental-information.

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
