# Peer review of "The dopamine receptor D5 gene shows signs of independent erosion in toothed and baleen whales"

_PeerJ, doi:10.7717/peerj.7758_

## Round 0.1 · original submission · Major Revisions

I have now received reviews from two anonymous referees. I concur with their overall assessment, which is that the conclusions are sound but the writing and other aspects of presentation (especially figures) could use some improvement.

There are a few things I would like to highlight:

1. I would like the authors to make sure they are using gene names that conform to the HUGO names (those from https://www.genenames.org/), as much as possible. Obviously, the HUGO names are for human genes but the authors should use the human name for orthologs of those genes unless there is a clear consensus in the literature supporting the use of another name. I am concerned about this because I noticed that the Bos taurus genome has SLC2A9 immediately downstream of DRD5. I did not see this in their Fig. 1. If there are any "missing" genes in the map the authors should be very clear about why they are omitted from their gene maps.

2. There is some terminology (leftwards and rightwards) that is unacceptable. As reviewer 2 suggests, these could be changed to 5' and 3'. I actually prefer "upstream" and "downstream", but I would also like to make two suggestions. Make sure that "upstream" refers to genes 5' of the start codon and "downstream" to genes that are after the stop codon (i.e., "...we used CPEB2 as an anchoring gene to search the for genes downstream of DRD5...")

3. Suggestions to clean up the writing:

a. line 141 - the authors state "...10 species presenting DRD5 annotation tagged as ‘low-quality protein’ producing gene," This makes it sound like the gene produces a low-quality protein. This low-quality designation refers to issues that are likely to be sequencing errors. The authors should acknowledge the fact that the issue for low-quality proteins is the quality of the genome sequence, not the quality of the protein product.

b. Make sure italics are used appropriately throughout.

c. Fig. 2 - The authors state that "Unknown regions, either resulting from genome poor assembly or coverage (Ns in B. a. scammoni) or due to small genomic scaffold size (B. bonaerensis), are represented by black regions." First, the proper word order would be "poor genome assembly", though I would suggest "low-quality genome assembly". In a sense, all of these things are poor genome assembly. Overall, the authors should be very careful to make sure readers understand that the issue is the assembly quality.

It might even be a good idea to acknowledge somewhere in the paper that improved genome assembly make this type of analysis can be misled by low-quality genome assemblies. I find one of the best arguments for gene loss is that many members of a specific clade have genes that appear to be pseudogenes.

I have only highlighted a few cases. The authors should take care to make sure the language is clear.

Finally, I would like the authors to add a paragraph to the introduction placing the concept of gene loss/pseudogenization in a broader framework. Reviewer 2 provides two citations. I'd like to point the authors to an older paper of mine:

Braun, E. L. (2003). Innovation from reduction: gene loss, domain loss and sequence divergence in genome evolution. Applied bioinformatics, 2, 13-34. ---> A pdf is available from https://people.clas.ufl.edu/ebraun/files/ApplBioinf-2003-Braun-13-34.pdf

I look forward to a revised version of this manuscript.

Reviewer 1 ·

Basic reporting

The writing is solid and easy to follow. References are adequate as is. The structure of the paper is simple and fine as is. Additional analyses would benefit the paper to further corroborate the evidence of gene inactivation or reduction in selection pressure (see below).

Experimental design

I think additional analyses should be done, but the analyses presented seem fine for documenting and confirming inactivating mutations in this gene across Mammalia.

Validity of the findings

By its nature, the inference on function in the paper is speculative, but the authors do not overstate the significance of their results.

Additional comments

Alves et al. examine inactivating mutations in the DRD5 gene, a dopamine receptor in mammals. I am not an expert on dopamine pathways, but can comment on the comparative analyses done on various published mammalian genomes. The authors pulled out DRD5 gene sequences from more than 100 mammal species, and describe sequences from taxa that show indications that this gene is not functional (frameshifts, stop codons, etc.). They find that cetaceans, both mysticetes and odontocetes, are incapacitated along with the golden mole (Chrysochloris). The authors suggest that these mammals are therefore naturally occurring knock outs for the DRD5 gene and could be used as models for the study of vasoconstriction. The authors use synteny relationships and various other procedures to compile the published data and map mutations that disrupt the reading frame in various mammalian taxa. The analyses are sound and the authors do not overstate conclusions, but the results are a speculative, involve only one gene KO, and the work will not overwhelm any reader I think. Some additional analyses should be done to give the limited work done here more weight given that no new data were described and that the inferences remain pretty speculative (see below).

Things that the reviewers might consider in improving their manuscript are listed below:
1) In the Introduction, the authors note that DRD5 knockouts have effects on renal salt balance. Is this of importance for cetaceans that live in saltwater and is this related to hypertension?
2) In the Materials and Methods, the authors use single letter genus abbreviations for cetacean species. These genus names should be spelled out here as many readers will not be familiar with these genera, and the reader must guess as to which species are described because of these abbreviations?
3) line 91 typo; change “flaking” to “flanking”.
4) line 107; “a reference phylogenetic close 
species” is a bit awkward. Reword this? 
 There is additional awkward phrasing in this section that should be fixed.
5) For Odontoceti, all species but Physeter have a shared inactivating mutation, and all species for which data are coherent show evidence of KOs. For Mysticeti, there is a common inactivating mutation for all baleen whale species examined. The authors note that, “Importantly, no conserved 
mutations were detected between Odontoceti and Mysticeti clades, suggesting that DRD5 pseudogenization events occurred independently after the divergence of both lineages (Figure 2)“. The inactivating mutations alone suggest independent inactivation in three cetacean lineages: mysticetes, Physeter, and all remaining odontocetes (except Physeter). However, the authors should try to corroborate this result by reconstructing dN/dS on their gene tree for DRD5. What is the ancestral dN/dS for this gene where the gene is presumably functional? For the authors interpretation, the stem cetacean branch (common ancestral branch) should have a dN/dS that suggests a functional gene with adaptive constraints (i.e., dN/dS much less than the neutral dN/dS of 1.0). Is this the case, or is the dN/dS on this stem cetacean branch much higher than the ancestral dN/dS for this gene in mammals that have functional copies of this gene. Also, the same question could be asked for the stem Odontoceti and stem Mysticeti branches. Does these have low dN/dS that is indicative of functional constraints (and suggesting independent KOs of the gene in Mysticeti and also in Odontoceti), or is dN/dS not significantly different from 1.0, suggesting neutral evolution that expected following a gene inactivation? These analyses should be done to bolster the authors’ conclusions about where these gene KO events occurred, especially considering that no new data were collected for the paper, that limited analyses were done, and that conclusions of the paper are already fairly speculative regarding functional outcomes for cetaceans given the gene KOs inferred. Basically, is there further evidence from dN/dS that DRD5 was KOed two or more times within Cetacea, or do dN/dS instead suggest a single KO of the gene on the stem to Cetacea with lags in mutational evidence that the gene was inactivated (common in cetaceans with slow mutation rates) and perhaps evidence from regulatory regions of the gene that DRD5 is inactivated earlier?
6) line 225 and following paragraph. The authors again refer to genera of mammals by using abbreviated genus names. The average reader will have no idea what the authors are talking about unless these genus names are spelled out here.
7) The authors note DRD5 KO in one burrowing mammal but not many other burrowing or deep diving mammals. Do the authors see any large differences in selective constraints, perhaps summarized by dN/dS for these “hypoxia lineages” or changes in otherwise well-conserved amino acid positions for this gene. Ideally, the authors could compare dN/dS on these hypoxia lineages relative to the dN/dS on taxonomic lineages that are not characterized by deep diving or burrowing. For example, what is the dN/dS in mole rats, moles, and deep diving pinnipeds (e.g., elephant seal). Perhaps there are shifts in dN/dS in these lineages that suggests DRD5 KOs that show lags in accumulation of inactivating mutations. The authors could either estimate dN/dS for each branch of their DRD5 tree separately or otherwise group branches in different categories (deep diver vs. not; burrower vs. non-burrower) to look for effects. Because the inferences in the paper with regards to functional impacts are currently speculative, the authors should try to bolster their inferences by doing some additional analyses of the data that they have worked to compile for this short paper.
8) Figure 1. Genus names of species should again be spelled out.
9) Figure 3 may not be necessary for this short paper?

Reviewer 2 ·

Basic reporting

I think this manuscript needs some english editing, as some passages are difficult to understand. Additionally some terminology (e.g. rightwards) should be replaced with common jargon.

I believe that the authors should make a better effort to introduce the problem. The manuscript would benefit if the authors make a smooth transition between the first and second paragraph in the introduction. As is now, the first paragraph is mainly devoted to the physiology of dopamine receptors and a little bit of evolution, whereas in the second the authors state their goal(s) and result in (mostly in) cetaceans. Something in between would be acknowledged.

The structure of the article is ok. Figures need some rethinking, in my opinion, figures 1 and 3 could be reported as supplementary material and figure 2 needs to be remodeled.

This manuscript is a self-contained unit, but in my opinion could be shortened to a "short communication" format, mostly reporting the cetacean result.

Experimental design

The research proposed is within the aims and scope of the journal

The research question is well defined.

The methods are well described and sufficient to replicate

Validity of the findings

This research is presenting a novel case of gene loss in cetaceans (and other mammals). The functional relevance is hard to assess, but some speculation is written in the manuscript.

The data they collected is sufficient to conclude that some mammalian lineages have lost DRD5.

Given the mammalian lineages they are working on, the main result is the gene loss. Some speculation regarding the functional consequences related to not having DRD5 is presented. They also highlight that cetaceans represent a natural KO system, however, it is also true that very limited research can be performed in this group.

Additional comments

General comment
This paper is trying to understand the evolution of the DRD5 gene in mammals, with an emphasis in cetaceans. I believe that this paper should be shortened to a "short communication" format. Cases out of cetacea should be removed, or commented in a couple of lines, no more. Figure 1 and 3 should be reported as a supplementary figure, and figure 2 should be extensively modified.

specific comments
In the introduction, I think there is a sort of disconnection between the first paragraph, which is mainly devoted to the physiology of dopamine receptors and a little bit of evolution, and the second one in which they state their goal(s) and results in (mostly in) cetaceans. I think the manuscript would benefit if the authors make a smooth transition between the first and second paragraph. In the case of the results section, I believe that the first paragraph, which is mostly devoted to technical issues, should be shortened. The last two sections of this section seem to be a small detail, as most of the paper (including figures) is dedicated to cetaceans. Figures are not attractive; I think the authors should make an effort to improve them.

Line 92: I suggest changing rightwards by 3´ side.
Line 93: I suggest changing leftwards by 5´ side.
Lines 93 to 95: I do not understand what are they trying to say, maybe expressing the idea in a different way would improve the text.
Line 102: what is the meaning of LQ?
Line 133 to 134: The description of this procedure should be expanded.
Line 149: the comma after the word "available" I think should be removed.
Line 177: I believe that the manuscript should benefit if common names are preferably used instead of specific names.
Lines 242 to 243: There are other papers reporting gene loss in cetaceans and linking this phenomenon to the conquest of the aquatic environment.

For example:
https://www.biorxiv.org/content/10.1101/521617v1
https://bmcgenomics.biomedcentral.com/articles/10.1186/1471-2164-15-869

Figure 1 & 2: I suggest using common names.

Figure 1: Having the names of each syntenic gene in each species make the figure very busy. My suggestion would be to have the names of the syntenic genes only in Bos taurus, and join them using vertical lines.

Figure 2: This figure is not attractive, and it does not communicate its message. I strongly suggest that these results should be presented in a phylogenetic context. Additionally, the symbology used is difficult to follow. Maybe using bars, stars, triangles instead of just bars would maximize the message of the figure.

---

## Round 0.2 · Minor Revisions

I was waiting for two re-reviews and have, at this point, only received one. However, I decided that I should move forward for two reasons: first, I feel the re-review that I have is sufficiently positive that your manuscript warrants acceptance based on the science; second, I have communicated with the second re-reviewer and s/he had some unexpected pressures on his/her schedule that would mean we would have to wait an unreasonable amount of time. I do not want to do that do you.

I debated between "accept" and "minor revisions" and came down on the side of the latter because, despite the fact that I feel the science is sound, there are a number of edits that you should make and I'd like to read the manuscript to evaluate they way you have addressed them. Please use the annotated manuscript as a guide. Also, I have noticed some issues with Table 1 (the dN/dS analysis):

1. Make the N and S of dN/dS subscripts and make the d's italic.
2. Be consistent in your use of decimal points (you have one case where you use a comma rather than the standard American usage of a period, in other places you use a period - change the comma.
3. The phrasing of "Not statistically significantly ≠ from 1" and "statistically significantly

Reviewer 1 ·

Basic reporting

The authors have made improvements to the paper following the reviewers' and editor's guidance. Generally my main concerns were addressed. I have put comments and small suggested edits using track changes on top of the authors' edits and will submit this. Perhaps the authors could consider some of these suggested changes and adjustments.

Experimental design

Generally, the approach taken is valid. I would have like to have seen a bit more, but the paper is improved in revision by adding some dN/dS analyses.

Validity of the findings

Generally the results seem valid. dn/ds values should be reported for all categories. As currently written, I am maybe not sure that three independent KOs within Cetacea is supported, so I suggest that the authors consider this topic a bit more.

Additional comments

The authors have removed the manuscript in revision and I think with just a little bit more tweaking it will be ready for publication.

Annotated reviews are not available for download in order to protect the identity of reviewers who chose to remain anonymous.

---

## Author Rebuttal · Round 0.2

Dear Prof. Braun,

We re-submit here the revise version of our manuscript. We provide below a detailed response to the comments and suggestions raised by yourself and the reviewers. We take the opportunity to thank you and the anonymous reviewers for the time and consideration in the revision of our work.

Sincerely,

Filipe Castro

(on behalf of all co-authors)

**The dopamine receptor $D_5$ gene shows signs of independent erosion in toothed and baleen whales**

**Answers to editor's comments**

**1. I would like the authors to make sure they are using gene names that conform to the HUGO names (those from https://www.genenames.org/), as much as possible. Obviously, the HUGO names are for human genes but the authors should use the human name for orthologs of those genes unless there is a clear consensus in the literature supporting the use of another name. I am concerned about this because I noticed that the Bos taurus genome has SLC2A9 immediately downstream of DRD5. I did not see this in their Fig. 1. If there are any "missing" genes in the map the authors should be very clear about why they are omitted from their gene maps.**

::. We confirm that all mentioned genes in the manuscript are in conformance with the HUGO nomenclature. We understand your concern regarding the missing SLC2A9 representation in the synteny maps. In the first submitted version of the manuscript we inadvertently failed to mention, in the material and methods section, that synteny maps were built considering the upstream and downstream DRD5 neighboring genes from the **DRD5 strand only**. For this reason, genes that are present in the opposite DNA strand, such as SLC2A9, were not included in the synteny map depicted in the former Figure 1, now supplementary Figure 1.

**2. There is some terminology (leftwards and rightwards) that is unacceptable. As reviewer 2 suggests, these could be changed to 5' and 3'. I actually prefer "upstream" and "downstream", but I would also like to make two suggestions. Make sure that "upstream" refers to genes 5' of the start codon and "downstream" to genes that are after the stop codon (i.e., "...we used CPEB2 as an anchoring gene to search the for genes downstream of DRD5…")**

::. We followed your suggestion and replaced 'rightwards' for downstream and 'leftwards' for upstream. We also confirm that "upstream" refers to genes 5' of the start codon and "downstream" to genes that are after the stop codon.

**3. Suggestions to clean up the writing:**

**a. line 141 - the authors state "...10 species presenting DRD5 annotation tagged as 'low-quality protein' producing gene,"** This makes it sound like the gene produces a low-quality protein. This low-quality designation refers to issues that are likely to be sequencing errors. The authors should acknowledge the fact that the issue for low-quality proteins is the quality of the genome sequence, not the quality of the protein product.

::. The expression 'low-quality protein' producing gene was altered. A clarification was added, in the material and methods section, regarding the meaning of the low-quality protein tag in NCBI RefSeq sequences (lines 119-124).

**b. Make sure italics are used appropriately throughout.**

::. We confirm that italics are used appropriately throughout the manuscript.

c. Fig. 2 - The authors state that "Unknown regions, either resulting from genome poor assembly or coverage (Ns in *B. a. scammoni*) or due to small genomic scaffold size (*B. bonaerensis*), are represented by black regions." First, the proper word order would be "poor genome assembly", though I would suggest "low-quality genome assembly". In a sense, all of these things are poor genome assembly. Overall, the authors should be very careful to make sure readers understand that the issue is the assembly quality.

::. For clarity, the statement was replaced by 'Fragmented or incomplete genomic regions are represented by black regions'.

**It might even be a good idea to acknowledge somewhere in the paper that improved genome assembly make this type of analysis can be misled by low-quality genome assemblies. I find one of the best arguments for gene loss is that many members of a specific clade have genes that appear to be pseudogenes.**

::. We incorporated this reasoning to the discussion.

I have only highlighted a few cases. The authors should take care to make sure the language is clear. The manuscript was fully revised.

Finally, I would like the authors to add a paragraph to the introduction placing the concept of gene loss/pseudogenization in a broader framework. Reviewer 2 provides two citations. I'd like to point the authors to an older paper of mine:

Braun, E. L. (2003). Innovation from reduction: gene loss, domain loss and sequence divergence in genome evolution. Applied bioinformatics, 2, 13-34. ---> A pdf is available from https://people.clas.ufl.edu/ebraun/files/ApplBioinf-2003-Braun-13-34.pdf

::. We appreciate this and the additional suggestions and added all the suggested references. The concept of gene loss in the introduction was also slightly expanded as requested.

**Answers to reviewers' comments and suggestions**

**Reviewer 1**

**Comments for the author**

**1) In the Introduction, the authors note that DRD5 knockouts have effects on renal salt balance. Is this of importance for cetaceans that live in saltwater and is this related to hypertension?**

::. This is well raised by the reviewer. By regulating salt levels in blood, renal salt balance can influence blood pressure. Although DRD5 has been mostly studied in brain, some studies address the interaction of DRD5 and the renal receptor Angiotensin II type 1, known to increase renal salt import and hence blood pressure (cited in the introduction). In fact, both receptors seem to yield opposing phenotypes; thus, DRD5 KO could exacerbate the hypertensive effects of Angiotensin II type 1. This was briefly clarified in the introduction. If the function and modulation of Angiotensin II type 1 is conserved in Cetacea, the suggested loss of DRD5 could alter the renal salt balance. For saltwater cetaceans this might seem counterintuitive; yet, other physiological mechanisms should be taken into consideration regarding the overall salt balance, such as the observed high blood volume or the adaptive evolution of other genes that modulate water and salt balance (BMC Evol Biol. 2013; 13: 189. Published online 2013 Sep 9. doi: 10.1186/1471-2148-13-189).

**2) In the Materials and Methods, the authors use single letter genus abbreviations for cetacean species. These genus names should be spelled out here as many readers will not be familiar with these genera, and the reader must guess as to which species are described because of these abbreviations?**

::. Species names are in conformity with the the PeerJ author instructions https://peerj.com/about/author-instructions/, Species formatting section).

Example:
"The next time that species is mentioned, abbreviate the name (i.e., the first letter of the genus followed by a period and the species), unless:
    There are two species that belong to different genera that nevertheless start with the same letter (e.g., Leopardus pardalis, the ocelot, and Lynx canadensis, the Canada lynx). Do not abbreviate the genus name." Accordingly, only Tursiops, Jaculus, Sousa, Delphinapterus and Neophocaena can be abbreviated.
Additionally, to improve the readership range of the manuscript, species names are now accompanied by the corresponding common name (including figures).

**3) line 91 typo; change "flaking" to "flanking".**

::. This typo was corrected.

**4) line 107; "a reference phylogenetic close   species" is a bit awkward. Reword this?    There is additional awkward phrasing in this section that should be fixed.**

::. The Methods section was thoroughly revised in compliance with the recommendations made by both reviewers.

**5) For Odontoceti, all species but Physeter have a shared inactivating mutation, and all species for which data are coherent show evidence of KOs. For Mysticeti, there is a common inactivating mutation for all baleen whale species examined. The authors note that, "Importantly, no conserved   mutations were detected between Odontoceti and Mysticeti clades, suggesting that DRD5 pseudogenization events occurred independently after the**

divergence of both lineages (Figure 2)". The inactivating mutations alone suggest independent inactivation in three cetacean lineages: mysticetes, Physeter, and all remaining odontocetes (except Physeter). However, the authors should try to corroborate this result by reconstructing dN/dS on their gene tree for DRD5. What is the ancestral dN/dS for this gene where the gene is presumably functional? For the authors interpretation, the stem cetacean branch (common ancestral branch) should have a dN/dS that suggests a functional gene with adaptive constraints (i.e., dN/dS much less than the neutral dN/dS of 1.0). Is this the case, or is the dN/dS on this stem cetacean branch much higher than the ancestral dN/dS for this gene in mammals that have functional copies of this gene. Also, the same question could be asked for the stem Odontoceti and stem Mysticeti branches. Does these have low dN/dS that is indicative of functional constraints (and suggesting independent KOs of the gene in Mysticeti and also in Odontoceti), or is dN/dS not significantly different from 1.0, suggesting neutral evolution that expected following a gene inactivation? These analyses should be done to bolster the authors' conclusions about where these gene KO events occurred, especially considering that no new data were collected for the paper, that limited analyses were done, and that conclusions of the paper are already fairly speculative regarding functional outcomes for cetaceans given the gene KOs inferred. Basically, is there further evidence from dN/dS that DRD5 was KOed two or more times within Cetacea, or do dN/dS instead suggest a single KO of the gene on the stem to Cetacea with lags in mutational evidence that the gene was inactivated (common in cetaceans with slow mutation rates) and perhaps evidence from regulatory regions of the gene that DRD5 is inactivated earlier?

::. The suggested dN/dS analyses were done, increasing the robustness of the of DRD5 pseudogenization inference in the studied lineages, with results supporting the DRD5 mutational evidences, suggesting independent DRD5 inactivation phenomena: one in the *Mysticeti* extant ancestor, other occurring in the Odontoceti ancestor (excluding *Physeter catodon*, sperm whale) and finally, an independent DRD5 inactivation phenomenon in *Physeter catodon* (sperm whale). The manuscript has now two additional sections, one concerning Phylogenetic and dN/dS analyses methodology, and the other concerning dN/dS analyses results.

6) line 225 and following paragraph. The authors again refer to genera of mammals by using abbreviated genus names. The average reader will have no idea what the authors are talking about unless these genus names are spelled out here.

::. Please refer to comment 2).

7) The authors note DRD5 KO in one burrowing mammal but not many other burrowing or deep diving mammals. Do the authors see any large differences in selective constraints, perhaps summarized by dN/dS for these "hypoxia lineages" or changes in otherwise well-conserved amino acid positions for this gene. Ideally, the authors could compare dN/dS on these hypoxia lineages relative to the dN/dS on taxonomic lineages that are not characterized by deep diving or burrowing. For example, what is the dN/dS in mole rats, moles, and deep diving pinnipeds (e.g., elephant seal). Perhaps there are shifts in dN/dS in these lineages that suggests DRD5 KOs that show lags in accumulation of inactivating mutations. The authors could either estimate dN/dS for each branch of their DRD5 tree separately or otherwise group branches in different categories (deep diver vs. not; burrower vs. non-burrower) to look for effects. Because the inferences in the paper with regards to functional impacts are currently speculative, the authors should try to bolster their inferences by doing some additional analyses of the data that they have worked to compile for this short paper.

::. This is indeed an interesting issue raised by the reviewer. Since reviewer no. 2 suggests that 'cases out of Cetacea should be removed or commented in a couple of lines, no more', also

including the suggestion of shortening the manuscript, and given the main emphasis of the analysis centered in cetaceans, it seems reasonable to us to not include this analysis in this short manuscript.

**8) Figure 1. Genus names of species should again be spelled out.**

::. Genus names are now included in this figure (formerly Figure 1, now moved to supplementary material, Supplementary Figure 1, following reviewer no. 2 suggestion).

**9) Figure 3 may not be necessary for this short paper?**

::. We agree with your suggestion and Figure 3 has been excluded from the manuscript.

**Reviewer 2**

**Comments for the author**

**Line 92: I suggest changing rightwards by 3´ side.**

::. 'Rightwards' was replaced by downstream throughout the manuscript.

**Line 93: I suggest changing leftwards by 5´ side.**

'Leftwards' was replaced by upstream throughout the manuscript.

**Lines 93 to 95: I do not understand what are they trying to say, maybe expressing the idea in a different way would improve the text.**

::. The Methods section was thoroughly revised in compliance with the recommendations made by both reviewers.

**Line 102: what is the meaning of LQ?**

::. A specific paragraph has been added to the material and methods section of the revised manuscript to clarify the meaning of the low-quality protein tag regarding NCBI RefSeq sequences (lines 119-124).

**Line 133 to 134: The description of this procedure should be expanded.**

::. The procedure for validation of the identified mutations was expanded as requested.

**Line 149: the comma after the word "available" I think should be removed.**

::. The comma was removed.

Line 177: I believe that the manuscript should benefit if common names are preferably used instead of specific names.

::. We agree with your comment. To improve the readership range of the manuscript, species names are now accompanied by the corresponding common name (including figures).

**Lines 242 to 243: There are other papers reporting gene loss in cetaceans and linking this phenomenon to the conquest of the aquatic environment. For example:**

https://www.biorxiv.org/content/10.1101/521617v1

https://bmcgenomics.biomedcentral.com/articles/10.1186/1471-2164-15-869

::. These are indeed important references and were added to the revised manuscript.

**Figure 1 & 2: I suggest using common names.**

::. We included the common names of each species in both figures (formerly known as Figures 1 and 2, now changed to Supplementary Figure 1 and Figure 2, respectively).

**Figure 1: Having the names of each syntenic gene in each species make the figure very busy. My suggestion would be to have the names of the syntenic genes only in *Bos taurus*, and join them using vertical lines.**

::. We agree and followed your suggestion, Supplementary Figure 1 (formerly Figure 1) is now cleaner and uses of vertical lines to join orthologous genes across the different species, with each line representing a different reference *Bos taurus* DRD5 neighboring gene.

Figure 2: This figure is not attractive, and it does not communicate its message. I strongly suggest that these results should be presented in a phylogenetic context. Additionally, the symbology used is difficult to follow. Maybe using bars, stars, triangles instead of just bars would maximize the message of the figure.

::. We agree and followed your suggestion of incorporating a phylogenetic context into the formerly Figure 2 (now Figure 1). We used the produced maximum likelihood (ML) phylogenetic tree regarding the studied Cetacean; *Hippopotamus amphibius* (hippopotamus), *Bos taurus* (cattle) and *Homo sapiens* (human) DRD5 coding sequences used in the dN/dS analyses for producing a cladogram enriched with Cetacean DRD5 mutational evidences mapped to the corresponding branches, with each branch color coded according to the corresponding DRD5 functionality status (functional or pseudogenized).

---

## Round 0.3 · accepted · Accept

I have evaluated all of your revisions and am very happy to accept the manuscript. Thank you for the careful attention you paid in the revision of several oddly phrased (but technically correct) statements (e.g., the revision of the dN/dS tests, "allelic pseudogenization", etc).

I look forward to seeing the final publication. Make sure to publicize it via twitter (and encourage your colleagues to do so as well!)